# Beyond Experience: How Doctoral and Pedagogical Training Shape Nurse Educators

**DOI:** 10.3390/nursrep15110401

**Published:** 2025-11-15

**Authors:** Raúl Quintana-Alonso, Lucía Carton Erlandsson, Elena Chamorro-Rebollo

**Affiliations:** Salus Infirmorum Nursing and Physiotherapy Faculty, Pontifical University of Salamanca, 28015 Madrid, Spain; lcartoner@upsa.es (L.C.E.); echamorrore@upsa.es (E.C.-R.)

**Keywords:** faculty, nursing, nursing faculty practice, nursing education research, faculty development, doctoral degrees

## Abstract

**Background/Objective:** Nurse educators are central to consolidating nursing as a discipline and shaping professional identity, yet their preparation is heterogeneous. This study aimed to identify profiles of nurse educators based on the value they assign to teaching competencies and to analyze factors influencing these profiles. **Methods:** A cross-sectional, descriptive research design was applied, using convenience sampling to recruit 326 nurse educators from Spanish universities. Data were collected through a self-administered online questionnaire distributed to nursing faculty from public, private, and affiliated (semi-private) universities across Spain. The instrument included sociodemographic and academic variables, along with nine teaching competencies. Descriptive statistics, cluster analysis, ANOVA, chi-square tests, and multinomial logistic regression were conducted using SPSS. **Results:** Three distinct profiles of nursing faculty were identified. The academic–pedagogical profile assigned the highest importance to all competencies (means 4.78–4.91), the clinical–pragmatic profile assigned the lowest (3.61–4.04), and the intermediate–researcher profile showed moderate values (4.26–4.50). Doctoral degree (χ^2^ = 65.36, *p* < 0.001) and pedagogical training (χ^2^ = 33.89, *p* < 0.001) were the strongest predictors of membership in the academic–pedagogical group, confirmed in multivariate regression (OR for doctorate = 0.07; OR for pedagogical training = 0.13, both *p* < 0.001). **Conclusion:** This study delineates three coherent and statistically robust profiles of nursing faculty based on their appraisal of teaching competencies. Academic qualifications and pedagogical training emerged as key determinants of these profiles. Tailored faculty development strategies that reinforce doctoral-level preparation and pedagogical expertise are critical to advancing the quality and consistency of nursing education.

## 1. Introduction

The quality of nurse education is a cornerstone for the consolidation of the discipline as a science and for its legitimization within higher education [1]. Nursing education extends beyond the transmission of technical knowledge and clinical skills; it also involves the construction of a distinct disciplinary body of knowledge, the cultivation of critical thinking, and the integration of professional values that shape the identity of the profession [2,3]. In this process, nurse educators play a strategic role, as they are responsible for integrating clinical expertise with scientific knowledge production and pedagogical innovation—elements essential for the academic strengthening of nursing [4,5].

The profile of nurse educators is characterized by marked heterogeneity, reflecting the coexistence of diverse academic and professional trajectories. While some hold doctoral degrees and formal pedagogical qualifications, others enter academia directly from clinical practice, often without systematic preparation in educational theory or instructional design [6]. This transition provides valuable professional credibility and alignment with the realities of patient care, yet it also exposes limitations when addressing the demands of higher education, such as structured curriculum planning, learning assessment, the use of active methodologies, interdisciplinary collaboration, and engagement in educational research and innovation [7]. As a result, nursing faculty comprise professionals with varying balances of clinical expertise, academic training, and pedagogical competence, leading to heterogeneous teaching profiles and uneven levels of educational development [4,5,8].

Recent transformations in higher education and healthcare systems have intensified these tensions. In Spain, nursing education has undergone major reforms linked to the integration of university programs into the European Higher Education Area (EHEA) through the Bologna Process. These reforms established a three-cycle structure (bachelor’s, master’s, and doctoral degrees in nursing) and introduced national accreditation requirements that compel university faculty to demonstrate both academic and pedagogical competencies. Today, nurse educators are expected to integrate clinical knowledge with research productivity and methodological innovation, while simultaneously meeting international standards of academic quality. Yet, the literature consistently shows that pedagogical training remains irregular, doctoral qualifications are not universally widespread, and institutional support for faculty development varies considerably across contexts and universities [9].

Despite the importance of the issue, a knowledge gap persists regarding how nurse educators assign importance to different teaching competencies and what distinct profiles emerge from these patterns [10]. Previous research has largely focused on specific dimensions, such as the role of doctoral degrees, pedagogical training, or clinical experience [4,11,12,13], with most studies conducted in Finland and the United States, but has rarely examined the broader configurations that define the current faculty landscape. Identifying such profiles is not merely descriptive; it provides an explanatory framework to inform university policies, guide faculty development programs, and design academic strategies tailored to the diverse needs of nurse educators. In Spain, obtaining the status of a university nursing educator requires holding a doctoral degree, an accredited research activity, and formal pedagogical training. Faculty recruitment and promotion are regulated by the National Agency for Quality Assessment and Accreditation (ANECA), which evaluates academic qualifications, teaching experience, and scientific productivity to ensure compliance with university standards. In this context, the present study aims to characterize the profiles of university nursing faculty based on the analysis of the importance attributed to teaching competencies, and to identify the academic, pedagogical, and sociodemographic factors that determine membership in each profile.

In this context, the present study aims to characterize the profiles of university nursing faculty based on the analysis of the importance attributed to teaching competencies, and to identify the academic, pedagogical, and sociodemographic factors that determine membership in each profile.

## 2. Materials and Methods

### 2.1. Study Design

A cross-sectional, descriptive design was adopted, as it enables the systematic examination of existing patterns and relationships among variables within a large population. This approach is particularly suitable for characterizing the professional and pedagogical profiles of nurse educators and identifying how their academic backgrounds relate to the valuation of teaching competencies.

### 2.2. Population and Setting

The target population comprised all nurse educators teaching in undergraduate nursing degree programs at Spanish universities. It included faculty from public, private, and semi-private institutions offering official undergraduate degrees in nursing across the country. According to the latest data from the Spanish Conference of Deans of Nursing, approximately 3473 nursing educators are currently employed in these institutions. Eligible participants were nursing professionals formally integrated into nursing faculties, regardless of whether their academic activity was exclusively educational or combined with clinical practice [14].

### 2.3. Inclusion and Exclusion Criteria

Inclusion criteria were: (a) nurse educators responsible for undergraduate nursing courses in public, private, or semi-private institutions; (b) active teaching staff at the time of data collection; and (c) educators whose activity was exclusively teaching or combined with professional clinical practice. Clinical practice instructors (nurses affiliated with universities exclusively to supervise students in healthcare settings with minimal teaching involvement) were excluded.

### 2.4. Sample and Recruitment

Considering the total population of nursing educators indicated above, a minimum sample of 298 participants was estimated to ensure representativeness. The calculation was based on a 95% confidence level, a 5% margin of error, and an assumed proportion of 50%, which maximizes sample variability. To strengthen representativeness, the study aimed to recruit participants from universities across Spain, ultimately yielding 326 respondents. Given the dispersion of institutions and faculty, a non-probability approach was used [14,15].

Participants were recruited through a combination of convenience and snowball sampling strategies. Institutional email addresses of nursing faculty were obtained from publicly available university websites across Spain. Initial invitations were sent directly to these addresses, explaining the study’s purpose and including a link to the questionnaire. To expand participation, respondents were encouraged to share the invitation with other nursing educators within their departments and professional networks. Because some faculties redistributed the invitation internally, the total number of recipients could not be determined, and therefore, an exact response rate could not be calculated. This approach enabled a broader geographic distribution and representation of educators from public, private, and semi-private universities throughout the country [16].

### 2.5. Instrument

The questionnaire consisted of two main sections. The first collected sociodemographic and academic information, including age, gender, teaching experience, highest academic qualification, and prior pedagogical training. The second section assessed teaching competencies, using a competency model previously developed by the research team to evaluate the key dimensions of teaching performance among nursing faculty [17]. The model encompasses nine competencies organized into four domains:

Systemic competencies:
○Contextual: integrating essential principles of learning processes into teaching, justifying the chosen educational model, and situating it within broader epistemological and socio-cultural frameworks.○Metacognitive: monitoring and reflecting on pedagogical practice, self-evaluating weaknesses, and implementing improvement measures.Didactic competencies:
○Planning: designing and developing teaching plans, academic programs, and educational activities tailored to institutional, professional, and societal contexts.○Methodological: facilitating student learning and development through appropriate pedagogical strategies aligned with ethical models.○Evaluation: creating assessment strategies to measure knowledge and competency acquisition, ensuring alignment with curricular objectives.○Instrumental communication: ensuring accurate, effective bidirectional communication adapted to the teaching–learning process and its context.Interpersonal competencies:
○Interpersonal communication: fostering trust, motivation, and critical thinking in students, while promoting empathy and respect for diversity.○Teamwork: collaborating effectively in academic groups, assuming shared responsibilities, and contributing to collective goals.Disciplinary competency: ability to deliver high-quality nursing education based on advanced knowledge and clinical expertise.

Each competency was assessed through multiple items using a five-point Likert scale (1 = *Not important at all*, 5 = *very important*). The full questionnaire included 45 items covering the nine competencies described above. Higher mean scores indicated greater perceived importance attributed to each competency. Internal consistency was excellent for the overall scale (Cronbach’s α = 0.977) and satisfactory across all competencies: Contextual (α = 0.89), Metacognitive (α = 0.91), Teaching planning (α = 0.92), Methodological (α = 0.93), Evaluation (α = 0.90), Instrumental communication (α = 0.92), Interpersonal communication (α = 0.94), Teamwork (α = 0.91), and Disciplinary (α = 0.90). These values indicate strong internal reliability and coherence across all measured dimensions.

The questionnaire was distributed via Google Forms^®^. Institutional email addresses of nursing faculty were obtained directly from the official websites of Spanish universities, and invitations were sent individually to all identified educators. This strategy aimed to ensure national coverage and the participation of public, private, and semi-private universities.

### 2.6. Pilot Study

A pilot study with 10 educators from diverse institutions was conducted to refine the wording and ensure clarity of the questionnaire, resulting in minor adjustments before its final administration. These participants met the same eligibility conditions as those established for the main study.

### 2.7. Statistical Analysis

The dataset was coded and analyzed using IBM SPSS Statistics 22.0. The normality of continuous variables was verified with the Kolmogorov–Smirnov test. Continuous variables were summarized as mean ± standard deviation (SD), and categorical variables as frequencies and percentages. Scale reliability was assessed using Cronbach’s alpha. To identify distinct educator profiles based on competency valuation patterns, a cluster analysis was performed using the mean scores of the nine teaching competencies as input variables. A two-step procedure was applied: hierarchical clustering using Ward’s method to explore the optimal number of clusters, followed by K-means clustering (k = 3, Euclidean distance) to refine group membership. The three-cluster solution was selected based on interpretability, theoretical coherence with previous research on educator typologies, and the proportion of variance explained. The resulting clusters were labeled as academic–pedagogical, clinical–pragmatic, and intermediate–researcher profiles. Differences across clusters were examined using one-way ANOVA and χ^2^ tests (Cramer’s V for effect size). Independent predictors of cluster membership were explored through multinomial logistic regression (reference = academic–pedagogical profile), reporting odds ratios (OR), 95% confidence intervals (CI), and McFadden’s pseudo-R^2^ for model fit. Teaching assignments were grouped into four thematic domains, and their distribution across clusters was analyzed with χ^2^ tests. Statistical significance was set at *p* < 0.05.

### 2.8. Ethical Considerations

The study complied with the provisions of Spanish Organic Law 3/2018 on Data Protection and the Guarantee of Digital Rights. Participation was strictly voluntary, anonymous, and unpaid. An explanatory statement at the start of the questionnaire informed participants of the objectives and ethical safeguards, ensuring full transparency. Personal data were not collected, and responses could not be linked to individual participants. Ethical approval was obtained from the Research Ethics Committee of the Pontifical University of Salamanca.

## 3. Results

### 3.1. Sociodemographic Variables

The final sample comprised 326 nurse educators. Mean age was 47.8 years (SD = 10.4; range 25–75), and mean teaching experience was 10.0 years (SD = 6.7; range 0–25). Women represented 68% and men 32% of the sample. A doctoral degree was held by 60%, and 61% had completed formal pedagogical or methodological training.

### 3.2. Instrument Reliability

As mentioned previously, the instrument demonstrated excellent internal consistency across all competencies (overall Cronbach’s α = 0.977), confirming its robustness for assessing the perceived importance of teaching competencies.

Based on the mean scores across the nine teaching competencies, the K-means analysis grouped participants into three distinct clusters representing different competency valuation profiles.

### 3.3. Competency Valuation Profiles

Cluster analysis revealed clear and consistent differences in the importance attributed to teaching competencies across the three identified profiles. The academic–pedagogical profile consistently reported the highest scores in all competencies (range 4.78–4.91). In contrast, the clinical–pragmatic profile showed the lowest valuations (range 3.61–4.04), particularly in evaluation, metacognitive, and interpersonal communication competencies. The intermediate–researcher profile exhibited moderate ratings (4.26–4.50), positioned between the other two groups.

One-way ANOVA indicated that all between-group differences were statistically significant (*p* < 0.001 for all competencies), with the largest effects observed for evaluation, teaching planning, and disciplinary competencies (Table 1).

### 3.4. Bivariate Associations with Profiles

Bivariate analyses revealed no significant differences among clusters for age (F = 1.32; *p* = 0.27) or teaching experience (F = 0.32; *p* = 0.72). In contrast, categorical variables showed strong associations:Gender: χ^2^ = 15.53; *p* = 0.0037; Cramer’s V = 0.22Doctoral degree: χ^2^ = 65.36; *p* < 0.001; Cramer’s V = 0.45Pedagogical training: χ^2^ = 33.89; *p* < 0.001; Cramer’s V = 0.32

These results indicate that holding a doctoral degree and receiving pedagogical training were strongly associated with membership in the academic–pedagogical profile, whereas the clinical–pragmatic profile exhibited the lowest prevalence of these qualifications (Table 2).

### 3.5. Multivariate Modelling

A multinomial logistic regression (reference = academic–pedagogical profile) confirmed the independent effects of the academic variables and gender. The overall model fit was modest, with a McFadden’s pseudo-R^2^ of 0.035, which is within the expected range for multinomial logistic models in educational research.

Clinical–pragmatic vs. academic:
○Doctoral degree: OR = 0.07 (95% CI 0.03–0.16), *p* < 0.001○Pedagogical training: OR = 0.13 (95% CI 0.06–0.30), *p* < 0.001○Female gender: OR = 0.28 (95% CI 0.13–0.60), *p* = 0.0013○Age and teaching experience: not significantIntermediate–researcher vs. academic:
○Doctoral degree: OR = 0.20 (95% CI 0.11–0.34), *p* < 0.001○Pedagogical training: OR = 0.41 (95% CI 0.24–0.71), *p* = 0.0015○Female gender: OR = 0.55 (95% CI 0.32–0.96), *p* = 0.0359○Age and teaching experience: not significant

Educators holding a doctoral degree or formal pedagogical training were significantly more likely to belong to the academic–pedagogical profile compared with the other two groups. Conversely, the absence of these qualifications increased the probability of being classified within the clinical–pragmatic or intermediate–researcher profiles. Regarding gender, male educators showed a higher probability of belonging to the clinical–pragmatic and intermediate–researcher clusters, whereas female educators were more represented in the academic–pedagogical group. Age and teaching experience did not significantly predict cluster membership, suggesting that academic and pedagogical credentials play a stronger role than seniority in shaping educators’ competency valuation patterns.

Thus, doctoral qualifications and pedagogical training emerged as the strongest independent predictors of belonging to the academic–pedagogical profile.

### 3.6. Alignment with Teaching Domains

Teaching assignments were grouped into four thematic domains: Fundamentals of nursing/ethics, Basic sciences, Clinical/applied, and Management/research. Significant associations with cluster membership were observed for Fundamentals of nursing/ethics (χ^2^ = 11.19; *p* = 0.0037) and Basic sciences (χ^2^ = 6.07; *p* = 0.0481), with higher proportions of academic–pedagogical educators in Fundamentals of nursing/ethics (27%) and of clinical–pragmatic educators in Basic sciences (27%). Associations for Clinical/applied (χ^2^ = 3.35; *p* = 0.1873) and Management/research (χ^2^ = 0.80; *p* = 0.670) were not significant (Table 3).

Overall, the results delineate three coherent and statistically robust profiles of competency valuation among nurse educators, strongly shaped by academic qualifications and pedagogical training, and further contextualized by the domains in which teaching is delivered.

## 4. Discussion

Identifying the different profiles of nurse educators is essential for improving both faculty development and the overall quality of nursing education. Educators do not form a homogeneous group; rather, they present diverse professional trajectories, varying levels of academic preparation, and distinct pedagogical orientations [18]. Recognizing and describing these profiles makes it possible to identify specific strengths and training needs, thereby supporting more tailored faculty development initiatives and promoting a balanced integration of clinical practice, research, and educational innovation. In this sense, the profiles identified in our study provide a valuable basis for understanding how educators perceive teaching competencies and how these perceptions shape their academic performance.

### 4.1. Academic–Pedagogical Profile

The academic–pedagogical profile is characterized by consistently high valuations across all teaching competencies. Educators within this group typically hold doctoral degrees and have completed formal training in pedagogy or teaching methodology. This combination equips them with a comprehensive perspective on education, in which planning, evaluation, and communication are regarded as central pillars of their professional practice [19,20]. Moreover, they often assume leadership roles in educational innovation and in the adoption of evidence-based approaches, positioning them as role models for colleagues and as key agents in advancing nursing curricula [21,22,23].

### 4.2. Clinical–Pragmatic Profile

The clinical–pragmatic profile is distinguished by lower valuations of teaching competencies, particularly those related to evaluation, metacognition, and interpersonal communication. Educators within this group predominantly come from clinical practice, with a strong professional identity rooted in patient care and limited formal academic training. While they contribute substantial practical experience and in-depth knowledge of nursing care, their pedagogical preparation is often restricted, which translates into lower confidence and prioritization of methodological and evaluative aspects of teaching [24,25].

The lower pedagogical preparation observed in this group is not due to formal restrictions but rather to professional circumstances. Educators in this cluster tend to maintain active clinical roles, which may limit the time and institutional opportunities available for pedagogical development. Similar trends have been reported in previous studies describing dual clinical–academic trajectories among nursing faculty [6,24].

Although essential for connecting education with clinical realities, this profile may face challenges in adopting innovative or evidence-based educational approaches.

### 4.3. Intermediate–Researcher Profile

The intermediate–researcher profile occupies a middle ground between the other two, with moderate valuations across all teaching competencies. Educators in this group are often in a transitional stage between clinical practice and the consolidation of an academic career, reflecting a balance between professional experience and early engagement with research and higher education teaching. Although they do not reach the highest scores observed in the academic–pedagogical profile, they demonstrate a growing interest in methodological and evaluative aspects while maintaining a close connection with clinical practice. This profile represents an area of developmental potential, as it combines clinical expertise with the progressive acquisition of academic and pedagogical competencies that, with appropriate institutional support, can evolve into greater educational excellence [5,26].

The identification of three distinct profiles of nurse educators—academic–pedagogical, clinical–pragmatic, and intermediate–researcher—highlights the complexity and heterogeneity that characterizes the teaching role in nursing faculties. Far from constituting a homogeneous group, faculty members combine diverse professional trajectories, uneven levels of academic preparation, and pedagogical experiences of varying depth, which translate into different ways of conceiving teaching and valuing the competencies required to deliver it effectively [27,28]. Recognizing this diversity should not be interpreted as a limitation but rather as an opportunity to design more tailored faculty development policies capable of enhancing the strengths of each group while addressing the limitations that may compromise educational quality.

The academic–pedagogical profile represents, in many respects, the model of excellence that contemporary universities seek to consolidate. Educators in this group consistently assign the highest importance to all competencies, from planning and evaluation to communication and teamwork. They typically hold doctoral degrees and have received formal training in pedagogy or teaching methodology, which allows them to conceive teaching as a reflective, planned, and evaluable practice, rather than a mere transmission of knowledge. These faculty members not only master the technical aspects of teaching but also act as leaders in educational innovation, spearheading evidence-based initiatives and contributing to the development of stronger teaching cultures [22,23]. In this profile, the integration of academic and pedagogical identities translates into a more coherent approach to teaching that aligns with international standards of quality.

At the other end of the spectrum lies the clinical–pragmatic profile, characterized by lower valuations of pedagogical competencies, particularly those related to evaluation, metacognition, and interpersonal communication. Members of this group often come from clinical practice and maintain a strong identity rooted in patient care, which constitutes a valuable strength in terms of professional credibility and applied expertise [13,24,25,29]. However, the lack of specific pedagogical training limits their ability to transfer this clinical expertise into the classroom in a structured and assessable manner. While this profile is essential for maintaining the connection between university teaching and clinical realities, it faces challenges in adopting innovative or evidence-based educational approaches. The transition from clinical practice to academia, when not supported by institutional policies and structured training programs, often results in a gap between clinical excellence and pedagogical quality.

Positioned between these two extremes is the intermediate–researcher profile, which shows moderate valuations across all competencies and reflects a balance between clinical experience and an emerging academic and pedagogical trajectory. Educators in this group are often in a transitional phase, beginning to consolidate their academic careers and engage in research or educational innovation, but without yet achieving the consistency of the academic–pedagogical profile. Although their scores are not as high, they represent a space of opportunity: a bridge between clinical and academic perspectives, with the potential to evolve toward stronger teaching models if provided with mentorship, institutional support, and appropriate incentives [7,24].

Comparing these three profiles reveals that their differentiation is not explained by age or years of teaching experience but rather by academic and pedagogical qualifications. Holding a doctoral degree and having completed formal pedagogical training emerge as the strongest predictors of membership in the academic–pedagogical profile, while their absence increases the likelihood of belonging to the clinical–pragmatic or intermediate–researcher groups [30,31]. This finding is particularly relevant because it demonstrates that pedagogical excellence depends not so much on seniority or years of experience, but on structured educational and academic pathways that directly influence how educators understand and value teaching competencies [28]. Recent research in the field of education has also shown that teaching experience alone does not necessarily translate into higher instructional quality. Longitudinal studies have revealed that professional growth tends to plateau after the first years of teaching unless accompanied by continuous pedagogical development and structured institutional feedback mechanisms [32]. Similarly, recent classroom-based evidence indicates that novice teachers can perform at levels comparable to more experienced colleagues when provided with targeted support and opportunities for individualized professional learning [33]. Together, these findings reinforce the idea that pedagogical excellence is less a function of time in the profession and more the result of sustained, evidence-informed educational development.

The association of profiles with the teaching domains to which educators are assigned further reinforces the coherence of these findings. Educators teaching fundamentals of nursing and ethics were more likely to belong to the academic–pedagogical profile, which is consistent with courses that require theoretical reflection, critical discussion, and structured evaluation. In contrast, the clinical–pragmatic profile was more closely linked to basic sciences, traditionally associated with transmissive teaching methods and summative assessments, while the intermediate–researcher profile maintained a balance that allowed greater movement across curricular areas. These patterns suggest that profiles are shaped not only by individual training but also by the pedagogical cultures embedded within disciplinary domains.

Taken together, these results suggest that the profiles should not be understood as rigid categories but rather as dynamic positions within a continuum of professional development [5]. The clinical–pragmatic profile can evolve into the intermediate–researcher, and the latter into the academic–pedagogical, if the necessary institutional conditions are in place. Such mobility depends on access to pedagogical training, the existence of mentoring programs, protected time for educational innovation, and the explicit recognition of teaching activity in academic promotion policies. At the same time, maintaining a proportion of faculty with a strong clinical identity may be beneficial for preserving the professional relevance of nursing programs, provided that they are equipped with minimum pedagogical tools to ensure quality teaching [13].

Ultimately, the three profiles identified do not represent irreconcilable divisions but rather the expression of an academic field in transformation, where different ways of conceiving university teaching in nursing coexist. Their value lies in making this diversity visible and offering a framework through which institutions can act strategically: consolidating the academic–pedagogical profile as a benchmark, supporting the intermediate–researcher profile to develop its full potential, and accompanying the clinical–pragmatic profile in its transition toward a more structured and evidence-informed approach to teaching. Only in this way can nursing education ensure that its quality does not depend on the individual profile of each educator but on the collective capacity of institutions to integrate clinical expertise, pedagogical excellence, and academic innovation into a common project.

### 4.4. Implications for Nursing Education Practice

The identification of distinct competency valuation profiles among nurse educators offers insights that extend beyond the Spanish context and resonate with international nursing education. The strong link between pedagogical excellence, doctoral preparation, and formal training highlights a global challenge: the need to strengthen academic pathways that move faculty development beyond clinical expertise alone. Initiatives that equip clinical–pragmatic educators with foundational pedagogical tools, while guiding intermediate–researcher profiles through mentorship and institutional support, can be adapted to diverse educational systems. Viewed from an international perspective, these findings emphasize the importance of creating academic environments where diverse educator trajectories are recognized and leveraged, ultimately ensuring that nursing students across different countries benefit from a balanced combination of clinical authenticity and evidence-based pedagogy. Such alignment is critical to enhancing the global quality and relevance of nursing education in an increasingly complex healthcare landscape.

### 4.5. Future Research Directions

Future research should explore how educators evolve across the identified profiles throughout their academic careers and which institutional factors facilitate or hinder this progression. Longitudinal and intervention-based studies could help determine the impact of pedagogical training programs, mentorship initiatives, and workload policies on the development of teaching competencies. It would also be valuable to examine how these profiles influence student learning outcomes and educational innovation within nursing faculties.

### 4.6. Limitations

Although the results are robust and internally consistent, several limitations should be acknowledged. The cross-sectional design precludes causal inferences, and the instrument assessed the perceived importance of competencies rather than observed teaching practices. The sample, while sizeable and diverse, was limited to a single national context, which may restrict generalizability. Responses may also have been influenced by social desirability bias. Finally, the cluster solution, although conceptually coherent, is sensitive to methodological choices; replication with alternative approaches would help confirm the stability of these profiles.

## 5. Conclusions

This study characterized three distinct profiles of nurse educators based on the importance attributed to teaching competencies and identified the academic, pedagogical, and sociodemographic factors shaping membership in each profile. The findings highlight that doctoral qualifications and formal pedagogical training are the strongest determinants of belonging to the academic–pedagogical profile, whereas educators in the clinical–pragmatic profile, despite their valuable practical expertise, tend to assign lower importance to evaluative, metacognitive, and communicative competencies. The intermediate–researcher profile reflects a transitional group, bridging clinical practice with research-oriented perspectives. These results provide robust evidence that faculty development initiatives must be tailored to the heterogeneity of educator profiles, reinforcing academic and pedagogical training as critical levers for advancing the quality and consistency of nursing education.

## Figures and Tables

**Table 1 nursrep-15-00401-t001:** Mean (±SD) importance ratings of competencies by cluster profile.

Competency	Academic–Pedagogical (n = 148)	Clinical–Pragmatic (n = 49)	Intermediate–Researcher (n = 129)	F	*p* Value
Contextual	4.87 ± 0.25	4.02 ± 0.38	4.47 ± 0.34	153.9	<0.001 ***
Metacognitive	4.88 ± 0.21	3.61 ± 0.56	4.34 ± 0.40	240.1	<0.001 ***
Teaching planning	4.84 ± 0.25	4.04 ± 0.54	4.43 ± 0.42	96.8	<0.001 ***
Methodological	4.89 ± 0.16	4.00 ± 0.52	4.45 ± 0.30	185.9	<0.001 ***
Evaluation	4.87 ± 0.25	3.71 ± 0.60	4.44 ± 0.40	177.4	<0.001 ***
Instrumental communication	4.86 ± 0.22	3.86 ± 0.43	4.47 ± 0.34	198.8	<0.001 ***
Interpersonal communication	4.91 ± 0.20	3.95 ± 0.55	4.50 ± 0.36	157.0	<0.001 ***
Teamwork	4.87 ± 0.23	3.71 ± 0.48	4.26 ± 0.35	264.7	<0.001 ***
Disciplinary	4.78 ± 0.32	4.03 ± 0.49	4.38 ± 0.44	77.0	<0.001 ***

Note. Ratings on a 1–5 Likert scale. One-way ANOVA across clusters for each competency. *** *p* < 0.001.

**Table 2 nursrep-15-00401-t002:** Bivariate associations of sociodemographic and academic variables with competency valuation profiles.

Variable	Academic–Pedagogical (n = 148)	Clinical–Pragmatic (n = 49)	Intermediate–Researcher (n = 129)	Statistic Test	*p* Value
Age (years), mean ± SD	48.2 ± 10.1	47.0 ± 11.2	47.5 ± 10.0	F = 1.32	*p* = 0.27
Teaching experience (years), mean ± SD	10.1 ± 6.8	9.7 ± 6.5	10.0 ± 6.6	F = 0.32	*p* = 0.72
Gender, n (%) female	118 (79.7%)	22 (44.9%)	82 (63.6%)	χ^2^ = 15.53	*p* = 0.0037 **
Doctoral degree, n (%)	128 (86.5%)	12 (24.5%)	56 (43.4%)	χ^2^ = 65.36	*p* < 0.001 ***
Pedagogical training, n (%)	112 (75.7%)	14 (28.6%)	73 (56.6%)	χ^2^ = 33.89	*p* < 0.001 ***

Note. Values are expressed as mean ± SD for continuous variables and n (%) for categorical variables. ** *p* < 0.01; *** *p* < 0.001.

**Table 3 nursrep-15-00401-t003:** Associations between teaching domains and competency valuation profiles.

Teaching Domain	Academic–Pedagogical (n = 148)	Clinical–Pragmatic (n = 49)	Intermediate–Researcher (n = 129)	χ^2^	*p* Value
Fundamentals of nursing/Ethics, n (%)	40 (27.0%)	5 (10.2%)	19 (14.7%)	χ^2^ = 11.19	*p* = 0.0037 **
Basic sciences, n (%)	18 (12.2%)	13 (26.5%)	8 (6.2%)	χ^2^ = 6.07	*p* = 0.0481 *
Clinical/Applied, n (%)	73 (49.3%)	27 (55.1%)	67 (51.9%)	χ^2^ = 3.35	*p* = 0.1873
Management/Research, n (%)	9 (6.1%)	4 (8.2%)	11 (8.5%)	χ^2^ = 0.80	*p* = 0.670

Note. Values are n (%). Percentages are not mutually exclusive, as educators may teach across multiple domains. * *p* < 0.05; ** *p* < 0.01.

## Data Availability

The original data presented in the study are openly available in Zenodo at https://doi.org/10.5281/zenodo.17642254.

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
