# Peer review of "Beyond Experience: How Doctoral and Pedagogical Training Shape Nurse Educators"

_nursrep, 2025, doi:10.3390/nursrep15110401_

Round 1
Reviewer 1 Report
Comments and Suggestions for Authors
Dear authors,
The topic of the manuscript is highly relevant and relevant to the challenging question of how doctoral and pedagogical training shape nurse educators. The manuscript also provides important data on educator profiles and crucial implications for improving the practice of nursing education.
I would like to make suggestions for improving the manuscript:
The abstract is structured and contains a word count per the journal's guidelines for authors. I suggest that you indicate in line 14 whether all of the universities in Spain were included in the study and the exact number of universities. Please add this data.
Keywords: incomplete in relation to the title and content of the manuscript.
Introduction: The introduction is coherent and easy to follow.
Revisions are necessary for the following: In line 51, recent transformations are mentioned; however, this is a very general point of view. It is necessary to state where specifically and what changes occurred in the region/country where the study was conducted. At the same time, it would be significant to state the data from earlier studies in lines 60-61, as to when and in which countries they were conducted. It would be considerable to present data on the basic normative criteria for obtaining the status of a university nursing educator in Spain.
The Materials and Methods section is well designed with clearly separated subsections. It would be helpful to state that all of the universities in Spain were included in the study, as well as the exact number of universities. Is there any data on the number of nursing educators at these universities?
For a clearer understanding of the second section of the questionnaire, it would be desirable to present the statements based on which the competencies were assessed in the Appendix.
It would also be helpful to separate the pilot study from the published research more clearly. Also, what were the criteria for inclusion in this study? Were there any differences in their doctoral and pedagogical education? Are their data included in the presented results?
In the statistical analysis section, it is necessary to state the tests used to assess the normality of the data distribution.
The results are clear and concise. I suggest that the name of Table 1 should match the description given in lines 167-169.
Discussion. The discussion is extensive, and the authors draw attention to many important questions raised by their research and the results of other authors, through a logical discussion structure with adequate references from the literature.
The conclusions are concise, well-argued, and based on research results.
The references mentioned are relevant to the topic that the paper dealt with. Suggestion to check duplicate references, Mikkonen et al., 2018 paper cited twice (2 and 28).
I hope you find my comments helpful.
Author Response
Dear Editor,
We sincerely thank the reviewer for the careful reading and constructive feedback. Your comments have substantially improved the clarity, rigor, and readability of our manuscript. Below we respond point by point and indicate the corresponding changes.
1) Clarification regarding participating universities
Reviewer’s comment: Please specify whether all Spanish universities were included and provide the total number of institutions represented.
Response:
We have clarified this in Population and Setting. The questionnaire was distributed to nursing educators from all Spanish universities offering official undergraduate nursing degrees (public, private, and semi-private). To preserve confidentiality and avoid institutional bias, we did not collect the specific name of the university of each participant. Instead, we recorded the type of institution (public, private, semi-private), which is now explicitly reported in the manuscript. We also cite national figures from the Spanish Conference of Deans of Nursing (CNDE) indicating approximately 3,473 nursing educators nationwide.
2. Appendix with questionnaire items
Reviewer’s comment:
Include the items of the second section of the questionnaire as an appendix.
Response:
We appreciate this suggestion. Given that the questionnaire is extensive, including it as an appendix would significantly lengthen the manuscript. Therefore, following the journal’s recommendations for supplementary materials, we have decided to upload the full questionnaire as supplementary material, where all items are available for review.
3. Pilot study and inclusion criteria
Reviewer’s comment:
Please clarify the relationship between the pilot study and the main study, as well as the inclusion criteria for participants.
Response:
We have rewritten this subsection to clearly distinguish both phases. The revised text now specifies that the pilot study involved 10 educators selected under the same eligibility criteria as the main study. It reads:
“A pilot study involving 10 educators from diverse institutions was conducted to refine wording and ensure clarity, leading to minor adjustments prior to the final administration. These participants met the same eligibility conditions established for the main study.”
4. Statistical analysis and normality test
Reviewer’s comment:
Please specify how the normality of the variables was verified.
Response:
We have incorporated this information in the Statistical Analysis section, adding:
“The normality of continuous variables was verified using the Kolmogorov–Smirnov test.”
5) Table 1 title alignment with text
Reviewer’s comment: Ensure that the name of Table 1 matches the description in lines 167–169 (cluster analysis of competency ratings).
Response:
We updated the table title to explicitly reflect the clustering approach and content reported in the text.
6) Duplicate reference correction
Reviewer’s comment: Check for duplicate references (e.g., Mikkonen et al., 2018 listed twice).
Response:
We removed the duplicate Mikkonen et al., 2018 entry and renumbered the reference list accordingly to maintain Vancouver style consistency.
We would like to express our sincere appreciation to the reviewer for their constructive feedback and insightful suggestions. All revisions have been incorporated into the updated manuscript, with the corresponding changes highlighted in yellow to facilitate review.
Kind regards,
Raúl Quintana Alonso, PhD, RN.
Reviewer 2 Report
Comments and Suggestions for Authors
Thank you for leting me review your paper, which I found to be well written. There are a few minor points I think need addressing which I have listed here below:
Abstract: describes the method and need for study well, along with findings and suggestions
Introduction: good description of the need for this study, though if nurse educators involved are the ones lacking in pedagogical training, how will they know what they don't know when assigning importance to different teaching competencies?
Methods:
L120: 'This competency was evaluated only among nurse educators'; you said they were all nurse educators in (a) of your inclusion criteria.
L127: 'Invitations were sent through institutional email addresses' how did this work? Was it to course/programme emails and asked to be forwarded to the staff teaching on the courses? We need to know so we could reproduce the study from your description.
Results
L168 'competencies', not 'compe-tencies'
Otherwise well-displayed.
Discussion
Why is there a restriction on the clinical-pragmatic group's pedagogical preparation? Is it a lack of time/lack of availability of training?
The discussion is a bit repetitive.
Paragraph starting L313: have you found anything in the literature to back this up? What other research is there out there if any?
Any suggestions for future research?
References: please could they all be in English (see refs 9, 14, 30).
Author Response
Dear Editor,
We would like to sincerely thank the reviewer for the time and effort dedicated to carefully reading our manuscript and for providing valuable and constructive feedback. We are grateful for the insightful comments and suggestions, which have greatly contributed to improving the quality, clarity, and rigor of our work.
All the recommendations have been thoroughly addressed, and the manuscript has been revised accordingly. Below, we detail the specific changes made in response to each comment.
Reviewer comment:
Introduction: good description of the need for this study, though if nurse educators involved are the ones lacking in pedagogical training, how will they know what they don't know when assigning importance to different teaching competencies?
Author response:
We appreciate this valuable comment. The reviewer is correct in noting that educators with limited pedagogical preparation may have a restricted understanding of certain teaching competencies. Precisely for that reason, one of the study’s objectives was to capture these differences in perception. Our interest was not to assess actual pedagogical mastery but to identify how nurse educators, with diverse academic and professional backgrounds, perceive and prioritize the importance of teaching competencies.
This perspective is essential to understanding how varying degrees of pedagogical preparation shape educators’ conceptual frameworks and contribute to the emergence of distinct professional profiles. To clarify this rationale, we have refined the Introduction to make explicit that the study explores subjective valuation, acknowledging that perceived importance may be influenced by differences in pedagogical exposure.
Reviewer comment:
L120: “This competency was evaluated only among nurse educators”; you said they were all nurse educators in (a) of your inclusion criteria.
Author response:
We thank the reviewer for pointing out this inconsistency. This was a wording error in the original version. As stated in the inclusion criteria, all participants were nurse educators; therefore, the phrase suggesting that this competency was evaluated “only among nurse educators” was redundant. The sentence has been corrected to maintain internal consistency throughout the Methods section.
Reviewer comment:
L127: “Invitations were sent through institutional email addresses” — how did this work? Was it to course/programme emails and asked to be forwarded to the staff teaching on the courses? We need to know so we could reproduce the study from your description.
Author response:
We appreciate the reviewer’s attention to methodological transparency. The description has been clarified to specify how the invitation process was carried out. Institutional email addresses of nursing faculty were obtained directly from the official websites of the universities. The invitations were sent individually to these addresses through Google Forms®, ensuring national coverage and the inclusion of faculty from public, private, and semi-private (concerted) institutions.
Reviewer comment:
L168: “competencies”, not “compe-tencies”
Author response:
We thank the reviewer for noticing this typographical error. It has been corrected in the revised version.
Reviewer comment:
Why is there a restriction on the clinical-pragmatic group's pedagogical preparation? Is it a lack of time/lack of availability of training?
Author response:
We thank the reviewer for this insightful observation. The text has been clarified to explain that the lower pedagogical preparation observed in the clinical–pragmatic group is not due to formal restrictions but rather to professional circumstances. Many educators in this group continue to maintain active clinical roles, which often limit the time and institutional opportunities available for pedagogical training and professional development. This clarification has been added to the Discussion section.
Reviewer comment:
Paragraph starting L313: have you found anything in the literature to back this up? What other research is there out there if any?
Author response:
We appreciate the reviewer’s observation. The paragraph has been revised to clarify that the relationship between pedagogical excellence and structured educational pathways is consistent with findings reported in the broader field of teacher development. Empirical evidence in education supports the notion that teaching experience alone does not necessarily lead to greater instructional quality. Instead, sustained pedagogical training, mentoring, and institutional support are key factors for continued professional growth. This clarification has been added to the Discussion section to reinforce the consistency of our findings with the existing evidence base.
Reviewer comment:
Any suggestions for future research?
Author response:
We thank the reviewer for this helpful suggestion. A new subsection titled Future research directions has been added at the end of the Discussion to address this point explicitly. This section highlights the need for longitudinal and intervention-based studies to explore how educators progress across profiles and how institutional factors influence this evolution. It also suggests examining the impact of these profiles on student outcomes and educational innovation within nursing faculties.
Reviewer comment:
References: please could they all be in English (see refs 9, 14, 30).
Author response:
We appreciate the reviewer’s observation. All references originally written in Spanish have been translated into English while maintaining their original sources and URLs.
Once again, we would like to thank the reviewer for their thoughtful feedback and valuable insights. All modifications have been incorporated into the revised version of the manuscript, and the corresponding changes have been highlighted in yellow to facilitate the review process.
Kind regards,
Raúl Quintana Alonso, PhD, RN.